# Laryngeal Mask Airway Versus Endotracheal Intubation during Lacrimal Duct Stenosis Surgery in Children—A Retrospective Analysis

**DOI:** 10.3390/children11030320

**Published:** 2024-03-07

**Authors:** Nicolas Leister, Ludwig M. Heindl, Alexander C. Rokohl, Bernd W. Böttiger, Christoph Menzel, Christoph Ulrichs, Volker C. Schick

**Affiliations:** 1Department of Anesthesiology and Intensive Care Medicine, Faculty of Medicine, University Hospital of Cologne, University of Cologne, Kerpener Street 62, 50937 Cologne, Germany; bernd.boettiger@uk-koeln.de (B.W.B.); christoph.menzel@uk-koeln.de (C.M.); christoph.ulrichs@uk-koeln.de (C.U.); volker.schick@uk-koeln.de (V.C.S.); 2Department of Ophthalmology, Faculty of Medicine, University Hospital of Cologne, University of Cologne, Kerpener Street 62, 50937 Cologne, Germany; ludwig.heindl@uk-koeln.de (L.M.H.); alexander.rokohl@uk-koeln.de (A.C.R.)

**Keywords:** laryngeal mask airway, endotracheal intubation, pediatric anesthesia, pediatric ophthalmic surgery

## Abstract

*Background:* The use of laryngeal masks in the surgical treatment of infantile lacrimal duct stenosis is controversial due to the potential risk of aspiration. *Aims:* This study investigates airway procedures in children aged <6 years for surgery of lacrimal duct stenosis in a tertiary care university hospital. *Methods:* After institutional approval, airway procedures, duration of anesthesiological measures, and airway-related complications were retrospectively analyzed. Patients were divided into two groups according to the airway procedures used (endotracheal tube [ET] vs. laryngeal mask [LMA] airway). Associations were calculated using the Chi-square test or Mann-Whitney U-test. *Results:* Clinical data of 84 patients (ET *n* = 36 [42.9%] vs. LMA *n* = 48 [57.1%]) were analyzed. There were no significant differences in surgical treatment, age distribution, and pre-existing conditions between the groups. None of the patients showed evidence of tracheal aspiration or changes in measured oxygen saturation. LMA airway shortened time for anesthesia induction (*p* = 0.006) and time for recovery/emergence period (*p* = 0.03). In contrast, the time to discharge from the recovery room was significantly prolonged using LMA (*p* = 0.001). A total of 7 adverse events were recorded. Five of these were directly or indirectly related to ET (laryngo-/bronchospasm; muscle relaxant residual). *Conclusions:* LMA airway for infantile lacrimal duct stenosis seems to be a safe procedure and should be used in appropriate pediatric patients due to its lower invasiveness, low complication rate, and time savings.

## 1. What Is Already Known

-The use of laryngeal masks for airway management has become increasingly important in the field of pediatric anesthesia in recent years.

The use of laryngeal masks is still considered critical when performing surgery near the airway or when an increased risk of aspiration is suspected.

## 2. What Is New

-The use of laryngeal masks for probing, dilating, and irrigating lacrimal duct stenosis results in significantly shorter induction and emergence times in pediatric patients younger than 6 years of age.-Compared to endotracheal intubation, the use of laryngeal masks does not increase complications; some complications typically associated with endotracheal intubation could be avoided, despite secretions and the release of pus into the upper airway. Signs of relevant lower airway aspiration do not appear to occur.

## 3. Introduction

The use of a laryngeal mask airway, even in young children, has become standard in pediatric anesthesia and emergency medicine [1,2,3]. The major advantage is that a laryngeal mask insertion is less invasive than endotracheal intubation, thus avoiding irritation of the subglottic tracheobronchial system [4,5,6]. Utilization of the laryngeal mask airway is also recommended as an effective emergency procedure in the event of failed mask ventilation or intubation during difficult airway management, particularly in premature and newborn infants [7]. However, reduced aspiration protection is always an important consideration when planning airway management. This is especially important if the surgeon is actively introducing fluids into the airway or if bleeding and secretions are likely [4]. Airway management during short-term procedures such as probing, irrigating, and dilating the obstructed lacrimal duct in infants is also passed on this principle. Congenital obstruction of the lacrimal duct system is prevalent in approximately 6% of all newborns [8]. Failure of the nasolacrimal drainage system leads to overflow of tears and the risk of recurrent inflammation and chronicity [9]. Lacrimal duct obstruction is a common disease in early childhood. The prevalence varies from 5 to 20% during the first year of life [10]. Conservative therapy with monitoring, lacrimal sac massage, and antibiotics are common treatments [11]. If conservative therapy fails, probing the nasolacrimal duct is often a successful procedure for definitive therapy [12]. However, the most favorable timing for probing remains controversial [11]. The probing procedure requires treatment in the ophthalmology operating room under general anesthesia regardless of age. Despite the short procedure time, there is a theoretical risk of aspiration. The surgical technique, including irrigation and probing of the nasolacrimal duct, poses the risk of endotracheal aspiration of irrigation solution, blood, and/or pus [8]. An increased periprocedural risk in young children also contributes to the discussion about the appropriate timing of surgery. In particular, children under 3 years of age are at risk for respiratory complications, and anesthesiologists should avoid any airway interference, including endotracheal aspiration [13] For this reason, many anesthesiologists choose endotracheal intubation for airway protection during this procedure. Procedures using a laryngeal mask with continuous suctioning have been described in the literature [8]. To our knowledge, there is no general recommendation from the German or European Society of Anesthesiology.

At the authors’ institution, both endotracheal intubation and laryngeal masks are used for airway management in this surgical procedure; both procedures are used without continuous suctioning, based on the prevailing opinion at the authors’ hospital that this is more likely to cause mucosal lesions and that manipulation of the airway should be minimized in this way. In light of the above issues, airway management for this particular procedure is often a topic of discussion in the authors’ clinical routine. Mainly for at least three reasons: First, the surgical procedure is (usually) very short. Second, the risk of aspiration due to the underlying disease (pus) and the surgical technique (irrigation solution). Third, the population is at high risk for respiratory complications, especially children under the age of 3 years.

However, there are individual problems and risks associated with both procedures (endotracheal intubation or laryngeal mask airway). Objectification seems to be necessary in the absence of general recommendations.

This retrospective analysis was performed to evaluate the impact of each procedure on the course of general anesthesia, any complications, and procedural risks.

## 4. Materials and Methods

### 4.1. Ethics Committee Approval, In-Exclusion-Criteria and Endpoints

The responsible ethics committee (No: 22-1102; 24 August 2022) of the University Hospital of Cologne, Cologne, Germany approved the study and, also waived the requirement to obtain informed consent from patients to review and use the data. The study was conducted in accordance with the Helsinki Declaration on Patient Safety in Anesthesiology [14].

We retrospectively identified all children younger than 6 years of age who underwent lacrimal duct stenosis surgery during the corresponding period and analyzed their anesthesia protocols. The prevalence of patients with lacrimal duct stenosis younger than 6 years of age is 95.6% at our university hospital (regarding all patients younger than 18 years). As a tertiary center, we are only assigned cases that are refractory to conservative treatment.

The primary objective of this study was the analysis of the type of airway management (laryngeal mask airway (LMA) or endotracheal tube (ET)). Furthermore, we reviewed for procedure-related complications. In particular, indications of relevant aspiration events were sought: Description in the anesthesia record (e.g., subglottic secretions, gas exchange disturbance, need for excessive ventilatory support or supplemental oxygen), admission to the intensive/intermediate care unit during hospitalization, and readmission after discharge.

The secondary objective was to analyze the anesthetic approach (induction time: beginning of induction to surgery release by anesthesia team; emergence time: end of surgery to leaving operation room; recovery room time: start recovery room to recovery room discharge; anesthetic medication, and postoperative pain therapy) for probing, dilating, and irrigating the lacrimal duct in children younger than 6 years of age.

### 4.2. Patient and Data Selection

All children younger than 6 years who underwent lacrimal duct surgery under general anesthesia between January 2018 and February 2022 were included in this analysis. These children were all cared for by a final-year resident or specialist and supervised by a dedicated pediatric anesthesiologist in the Department of Anesthesiology at the University Hospital of Cologne. Anesthesiologists used standardized anesthesia protocols to document all data on vital signs and anesthetic process data. In the aforementioned anesthesia protocols used throughout the hospital, all analyzed events/vital signs were documented in a timeline and clearly timestamped (e.g., start of recovery room: 10:02; end of recovery room: 11:03; etc.). Patient data analysis included sex, age, length, weight, and American Society of Anesthesiologists physical status (ASA PS). All anesthetic medications, measures, and ventilation parameters were recorded. Adverse events and measures are documented according to established clinical standards. These data were retrospectively reviewed and analyzed for this study. Data collection was performed by two independent reviewers (NL, VCS). Nonconforming results were reevaluated.

### 4.3. General Anesthesia Administration and Documentation

At least one day before the scheduled surgery, a pre-anesthesia evaluation was performed by an anesthesiologist. All children were individually prepared according to in-house standards. In all cases, patients fasted according to the current international guidelines (until the end of 2021—meal: 6 h, milk: 4 h, clear liquids: 2 h; since 2022—meal 6 h, milk, and light meal: 4 h, breast milk: 3 h, clear liquids: 1 h) [15,16].

The decision about the type of airway management (LMA or ET) was made by the supervising pediatric anesthetist based on their experience/opinion with regard to the risk of aspiration associated with the surgical procedure. The airway procedure was pre-determined. Patients with an individual risk for/against one of the two procedures were excluded. There was no supervising anesthetist who used both procedures, i.e., either a laryngeal mask or endotracheal tube for all patients under care. The airway was established in all patients using a laryngeal mask (second generation) or endotracheal intubation (cuffed tubes). i.e., access was established during the induction period. Pressure-controlled ventilation with positive end-expiratory pressure (PEEP) was used. At the end of the surgery, the fluid in the nasopharynx was removed with a single suction. At the discretion of the anesthetist, the laryngeal mask or endotracheal tube was removed, and the patient was transferred to the recovery room. Patients were discharged from the recovery room to the regular ward after assessment by the specialist and the nurse in charge.

### 4.4. Performing Lacrimal Duct Surgery

The procedure included dilatation of the inferior lacrimal punctum as well as syringing and probing of the nasolacrimal duct. Initially, dilatation of the inferior lacrimal punctum was performed using a conical probe. A Bangerter lacrimal duct probe was then used for probing and syringing. After insertion of the Bangerter lacrimal probe into the inferior lacrimal punctum, the probe was slowly pushed forward until the surgeon reached a “hard stop” in the lacrimal fossa. Then, the probe was rotated 90 degrees and inserted into the nasolacrimal duct. During syringing, the probe was pushed slightly forward. During the procedure, between 5 and 15 mL of dyed Balanced Salt Solution (BSS) was injected. Consistency of the nasolacrimal duct was checked by staining a white Pro-ophta^®^ ocular stick to the nasopharynx with the dyed BSS.

### 4.5. Statistical Analysis

Demographic and sedation data were scanned into the electronic medical record (EMR) or recorded directly in the EMR. Data analyzed for this study were obtained from the EMR.

The statistical analysis was performed using SPSS version 29.0 (IBM, SPSS Statistics, IBM Corporation, Chicago, IL, USA). Distribution of demographic and medical data (age, weight, etc.) is presented as mean with ± SD (range). Descriptive data are presented as numbers and proportions. Associations were calculated using the Chi-square test and Mann-Whitney U-test. Significance was assumed when *p* ≤ 0.05.

## 5. Results

### 5.1. Demographic Data

Between January 2018 and February 2022, a total of *n* = 84 lacrimal duct surgeries were performed under general anesthesia in children younger than 6 years. Airway management was performed by endotracheal intubation (ET) in 36 children (42.9%) and by laryngeal mask (LMA) in 48 children (57.1%), allowing the creation of ET and LMA cohorts for retrospective analysis (cohort characteristics see Table 1). Final-year residents or junior specialists performed the airway procedure under the close supervision of six dedicated pediatric anesthetists, three of whom preferred LMA and three ET for the studied surgical procedure. The cohorts showed no significant differences in characteristics.

### 5.2. Anesthesia Performance

There were no significant differences between ET and LMA in the characteristics of individual anesthetic performances or anesthetic agents, except for opioids (*p* = 0.03) and for the muscle relaxant used for anesthesia (*p* = 0.01) (Table 2). Particularly, there was no significant difference in the mode of anesthesia induction and in anesthetics used for the maintenance of anesthesia.

### 5.3. Postoperative Pain Management

Prophylactic postoperative pain management (piritramide or nalbuphine and/or ibuprofen or paracetamol) showed no significant differences except for the intraoperative use of opioids for postoperative analgesia (*p* = 0.03) (Table 3): Patients in cohort LMA received significant more often parenteral opioid analgesics (*p* = 0.03) (piritramide or nalbuphine).

### 5.4. Oxygen and Saturation

Documented peripheral oxygen saturation without oxygen inhalation before induction of anesthesia showed no significant difference between cohorts (*p* = 0.74). The lowest intraoperative oxygen saturation (*p* = 0.77) and the first peripheral oxygen saturation measured after ET/LMA removal were without significant differences (*p* = 0.21). In addition, the lowest oxygen saturation during recovery (*p* = 0.91) and at discharge (*p* = 0.51) showed no significant differences. Oxygen fraction used during anesthesia was significantly lower in the LMA cohort compared to the ET cohort (*p* < 0.001) (Table 4). One child from cohort LMA received oxygen during the recovery room period (for no documented reason), but overall, there was no significant difference (*p* = 0.39) (Table 5).

### 5.5. Anesthesia Duration

The duration of anesthesia induction was significantly shorter in cohort LMA (*p* = 0.006), although one excessively long induction (duration 45 min due to difficult venous access) in cohort ET was excluded from the analysis. Duration from end of surgery to start of recovery room stay was significantly shorter in cohort LMA (*p* = 0.03), although two excessively long periods (41 and 45 min, without documented reason) in cohort ET were excluded from the analysis. Discharge to the regular ward was significantly later in cohort LMA (*p* = 0.001) (Table 6).

### 5.6. Adverse Events

A total of 7 adverse events were documented. Five complications were assigned to the ET cohort, and two complications were assigned to the LMA cohort. Airway-related complications (*n* = 4) were all attributed to the ET cohort. Residual neuromuscular block (Train-of-four (TOF) 3/4 at the end of surgery) after administration of muscle relaxants to induce anesthesia also occurred in the ET cohort. Two cases of postoperative emergence agitation were documented in the LMA cohort. For details, see Table 7. No patient had to be transferred to the intensive care unit (all patients could be transferred to the regular ward), nor did any patient have to be readmitted after discharge from the university hospital. All patients left the hospital the day after surgery.

## 6. Discussion

Airway management in children of all ages has undergone a paradigm shift in recent years from endotracheal intubation to laryngeal mask [3,17,18,19,20,21,22]. However, for many anesthesiologists, the use of a laryngeal mask still has certain implications for aspiration risk, particularly if the surgeon applies fluids to the airway during the procedure or if intraoperative access to the airway is restricted [23]. During probing, irrigating, and dilating of the lacrimal duct, this is exactly the risk: fluid is injected into the airway, and secretions such as pus may enter the airway due to dilation. In the literature, laryngeal mask use in dacryocystorhinostomy in adults is reported safely [24]. To the best of our knowledge, there is no reliable evidence on risk stratification in this area in children.

There is no expert consensus on definitive airway protection during pediatric lacrimal duct stenosis surgery. Even in the author’s medical center, there are different approaches. The team of dedicated pediatric anesthetists is actually divided into two cohorts: one group prefers endotracheal intubation, while others are comfortable with the use of a laryngeal mask airway. Under their supervision, the junior fellows performed the different airway procedures.

In the present study, the two airway procedures (ET vs. LMA) were retrospectively analyzed without significant differences. It was shown that the duration of anesthesia induction and recovery is significantly shorter with the use of a laryngeal mask airway. However, patients needed to stay in the recovery room significantly longer after using a laryngeal mask. This phase of the perioperative course of events can be influenced by many variables, particularly the significantly more frequent use of postoperative opioids in cohort LMA in this evaluation. The reason for this cannot be determined in the retrospective design.

The fact that the perioperative oxygen saturations at the respective time points are without significant differences demonstrates the comparable sufficiency of the applied airway procedures. However, it should be noted that the inspiratory oxygen fraction required to achieve these saturations was significantly lower in the LMA cohort. The airway-related adverse events reported were attributed to the method used for airway management (endotracheal intubation) [25]. Broncho- and laryngospasm, especially in hyper-reactive bronchial systems, may be typical complications of the use of an endotracheal tube inherent to the method. Muscle relaxants were only used to improve intubation conditions in the ET cohort; this group of drugs was not and is not generally used in the LMA airway [26,27,28].

The complications documented in the LMA cohort (emergence agitation) are not directly attributable to the method of airway management. Overall, the data presented here indicate that no relevant aspiration events (no description of any aspiration event in the anesthesia record (e.g., subglottic secretions), no gas exchange disturbance or need for ventilatory support, supplemental oxygen or admission to the intensive/intermediate care unit during hospitalization, no readmission after discharge) occurred in the LMA cohort, and thus the use of the laryngeal mask is possible and could be recommended, at least for this ophthalmic procedure. Continuous suctioning, as described in the literature, does not appear to be necessary to reduce the risk of aspiration [8]. In this way, the potential risk of mucosal lacerations from continuous upper airway manipulation could be reduced.

The adverse events documented in the present study, as well as the advantages in speed of induction and weaning from general anesthesia, support this conclusion.

## 7. Limitations

Our study was not without limitations. First, the current study is retrospective, and therefore, the retrospective design causes some limitations: The accuracy of the retrospective review of the handwritten medical records can be questioned. However, artifacts or true adverse events may be better distinguished or filtered out by manual documentation than by electronic documentation. Second, data analysis could be hampered by incomplete data sets. In addition, the evaluated time intervals should be considered as surrogate parameters for the effective induction and recovery time and may be influenced by various variables that cannot be determined retrospectively. Third, the regular pediatric anesthesia staff performed the procedures, not a single study physician. At the same time, diversity may offer advantages such as practical application and user-independent comparability.

## 8. Conclusions

Anesthesia induction (*p* = 0.006) and emergence (*p* = 0.03) were faster using LMA compared to ET. There was no evidence of increased risk of aspiration in those patients treated with LMA. Thus, LMA can be safely used for airway management in pediatric patients younger than 6 years for lacrimal duct surgery, including probing, dilating, and irrigating of the lacrimal duct.

## Figures and Tables

**Table 1 children-11-00320-t001:** Cohort characteristics.

	Cohort ET	Cohort LMA	*p*-Value
*n* (%)	36 (42.9)	48 (57.1)	
Age, months, mean ± SD (range)	25.2 ± 13.4 (4–57)	23.3 ± 12.5 (5–56)	0.49
1–11	8.5 ± 3.2 (4–11)	8.8 ± 1.9 (5–11)	
12–23	17.3 ± 2.9 (13–22)	17.4 ± 3.6 (12–23)	
24–35	29.8 ± 3.3 (24–35)	27.5 ± 2.5 (24–33)	
36–47	41.0 ± 3.9 (36–45)	41.3 ± 4.2 (37–47)	
48–59	55.0 ± 2.6 (52–57)	54.3 ± 1.5 (53–56)	
Female gender, *n* (%)	16 (44.4)	22 (45.8)	0.90
Body weight, kg, mean ± SD (range)	12.4 ± 2.6 (6–19)	12.3 ± 2.6 (8–19)	0.62
1–11	9.8 ± 3.1 (6–13)	9.2 ± 1.4 (8–12)	
12–23	11.5 ± 1.7 (8–15)	11.4 ± 1.5 (10–15)	
24–35	12.8 ± 1.5 (10–16)	13.6 ± 1.7 (11–17)	
36–47	14.9 ± 2.3 (13–18)	15.2 ± 2.6 (14–19)	
48–59	16.7 ± 2.1 (15–19)	16.3 ± 1.5 (15–18)	
Body size, cm, mean ± SD (range)	88.3 ± 9.1 (73–115)	88.5 ± 10.2 (68–110)	0.78
1–11	81 ± 8.5 (73–90)	74.7 ± 4.7 (68–80)	
12–23	82.5 ± 4.6 (73–90)	83.0 ± 5.1 (70–92)	
24–35	90.9 ± 4.6 (85–100)	93.5 ± 5.1 (84–100)	
36–47	99.0 ± 6.2 (91–106)	100.0 ± 3.8 (97–105)	
48–59	102.3 ± 11.2 (94–115)	107.7 ± 3.2 (104–110)	
ASA PS, *n* (%)			0.23
ASA PS I	31 (86.1)	46 (95.8)	
ASA PS II	4 (11.1)	2 (4.2)	
ASA PS III	1 (2.8)	0 (0)	

ET, endotracheal tube; LMA, laryngeal mask; SD, standard deviation; kg, kilogram; cm, centimeter; ASA PS, American Society of Anaesthesiologists Physical Status.

**Table 2 children-11-00320-t002:** Cohort anesthesia characteristics.

	Cohort ET	Cohort LMA	*p*-Value
Intravenous induction, *n* (%)	17 (47.2)	19 (39,6)	0.51
Inhalational induction, *n* (%)	19 (52,8)	29 (60.4)	
Volatile anesthetics used for maintenance, *n* (%)	34 (94.4)	47 (97.9)	0.57
Opioid used, *n* (%)			0.03
Remifentanil, *n* (%)	36 (100.0)	40 (83.3)	
Sufentanil, *n* (%)	0 (0.0)	5 (10.4)	
Fentanyl, *n* (%)	0 (0.0)	3 (6.3)	
Muscle Relaxants, *n* (%)	5 (13.9)	0 (0.0)	0.01

ET, endotracheal tube, LMA, and laryngeal mask.

**Table 3 children-11-00320-t003:** Postoperative pain management.

	Cohort ET	Cohort LMA	*p*-Value
Postoperative pain treatment, *n* (%)	33 (91.7)	44 (91.7)	1.00
Ibuprofen/Paracetamol, *n* (%)	33 (91.7)	41 (85.4)	0.38
Parenteral opioid analgesics, *n* (%)	8 (22.2)	22 (45.8)	0.03

ET, endotracheal tube, LMA, laryngeal mask.

**Table 4 children-11-00320-t004:** Oxygen fraction.

	Cohort ET	Cohort LMA	*p*-Value
FiO_2_ intraoperative lowest (mean ± SD (range))	0.47 ± 0.11 (0.30–0.70)	0.39 ± 0.11 (0.27–0.70)	<0.001

ET, endotracheal tube; LMA, laryngeal mask; FiO_2_, fraction of inspired oxygen; SD, standard deviation.

**Table 5 children-11-00320-t005:** Oxygen saturation.

	Cohort ET	Cohort LMA	*p*-Value
SpO_2_ Air (mean ± SD (range))	99.4 ± 1.1 (96–100)	99.4 ± 0.94 (96–100)	0.74
SpO_2_ intraoperative lowest (mean ± SD (range))	99.7 ± 1.2 (93–100)	99.8 ± 0.50 (98–100)	0.77
SpO_2_ postoperative first (mean ± SD (range))	97.5 ± 2.9 (85–100)	97.2 ± 2.0 (93–100)	0.21
SpO_2_ recovery room lowest (mean ± SD (range))	97.0 ± 2.0 (91–100)	97.0 ± 1.9 (90–100)	0.91
SpO_2_ recovery room discharge (mean ± SD (range))	98.3 ± 1.8 (91–100)	98.3 ± 1.5 (93–100)	0.51

ET, endotracheal tube; LMA, laryngeal mask; SpO_2_, peripheral capillary oxygen saturation; SD, standard deviation.

**Table 6 children-11-00320-t006:** Anesthesia times (minutes).

	Cohort ET	Cohort LMA	*p*-Value
Induction (mean ± SD (range))	14 ± 6 (4–30)	10 ± 5 (3–26)	0.006
Recovery/Emergence (mean ± SD (range))	14 ± 8 (0–37)	9 ± 6 (0–25)	0.03
Discharge recovery room (mean ± SD (range))	51 ± 17 (16–100)	74 ± 33 (16–150)	0.001

ET, endotracheal tube; LMA, laryngeal mask; SD, standard deviation.

**Table 7 children-11-00320-t007:** Adverse events.

No	Time	Cohort	Adverse Event	Management
1	postoperative	ET	residual neuromuscular block (TOF: 3/4)	reversal using acetylcholinesterase inhibitor
2	recovery room	ET	bronchial obstruction, respiratory distress	beta adrenergic drug application
3	postoperative	ET	laryngospasm after endotracheal tube removal	temporary desaturation, endotracheal re-intubation, prolonged wake-up
4	postoperative	ET	bronchial obstruction	beta adrenergic drug application, repeated propofol administration, ET removal and LMA insertion, prolonged wake-up
5	intraoperative	ET	airway loss, tube dislocation	temporary desaturation, endotracheal re-intubation
6	recovery room	LMA	emergence agitation	propofol administration in recovery room
7	recovery room	LMA	emergence agitation	propofol administration in recovery room

TOF, train-of-four; ET, endotracheal tube; LMA, laryngeal mask.

## Data Availability

The data presented in this study are available on request from the corresponding author. The data are not publicly available due to IRB decision. Parts of the data were shown as an abstract presentation at the 13th European Congress for Paediatric Anaesthesiology (Prague/28–30 September 2023).

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
