# Peer review of "Laryngeal Mask Airway Versus Endotracheal Intubation during Lacrimal Duct Stenosis Surgery in Children—A Retrospective Analysis"

_children, 2024, doi:10.3390/children11030320_

Round 1

Reviewer 1 Report

Comments and Suggestions for Authors

This simple and well-written paper presents a retrospective analysis of the use of LM airway for anesthesia during surgery for lacrimal stenosis in young children.  The authors appropriately justify the rationale for this study and adhere to proper ethical principles to perform this work.  Although this is not a randomized nor blinded trial, the size of the study population is significant, and the information in the manuscript sufficiently important to warrant consideration of the use of LM airways by other anesthesiologists, in this clinical setting.

The Abstract effectively and accurately summarizes the contents of the body text.

The Introduction is clear, well-referenced and succinct.  

The Methods provide adequate detail.  However, since this is not a randomized study, it would be useful to provide more information about the decision on which airway to use than what is given to the reader in lines 93 and 94.  In particular, do the authors have any information about whether some anesthesiologists usually (or always) use the LM versus the ET tube?  Or, do most anesthesiologists use either method, and make their choice based on characteristics of the individual patient, or after discussion of specifics of the case with the surgeon?  In both of these instances, there is the potential for significant confounding, but that would be more concerning if the latter pattern applies, because of the possibility that the LM might be used in lesions and there are a priori determined to present less risk.  Any information about the number of anesthesiologists involved, the number of cases by each and the distribution of airway types by anesthesiologist, would be useful in this regard.

The Discussion should also address the above point in some detail, irrespective of how much data are available on this issue.  It is inadequate to state that "this study has all the known limitations".

The conclusion is clear, direct, and justified based on the data presented.

Please correct misspellings:

In Table 1: “Femal” gender

In Table 3: “Postoperativ”

Author Response

Point-to-Point-Reviewer Response

Dear Editors & Reviewers

Thank you for having reviewed our manuscript. Please find enclosed the revised version of our manuscript entitled “Laryngeal mask airway versus endotracheal intubation during lacrimal duct stenosis surgery in children – a retrospective analysis”.

We are grateful for the reviewer´s invaluable comments and have adjusted the manuscript according to their recommendations. Please find below a point-by-point response, and attached the manuscript according to the editorial instructions. We hope that we have addressed all concerns appropriately and are looking forward to your decision.

Regards,

Nicolas Leister

on behalf of all co-authors

REVIEWER COMMENTS.

Reviewer #1

This simple and well-written paper presents a retrospective analysis of the use of LM airway for anesthesia during surgery for lacrimal stenosis in young children. The authors appropriately justify the rationale for this study and adhere to proper ethical principles to perform this work.  Although this is not a randomized nor blinded trial, the size of the study population is significant, and the information in the manuscript sufficiently important to warrant consideration of the use of LM airways by other anesthesiologists, in this clinical setting.

The Abstract effectively and accurately summarizes the contents of the body text.

We thank the reviewer.

The Introduction is clear, well-referenced and succinct. 

We thank the reviewer.

The Methods provide adequate detail.  However, since this is not a randomized study, it would be useful to provide more information about the decision on which airway to use than what is given to the reader in lines 93 and 94.  In particular, do the authors have any information about whether some anesthesiologists usually (or always) use the LM versus the ET tube?  Or, do most anesthesiologists use either method, and make their choice based on characteristics of the individual patient, or after discussion of specifics of the case with the surgeon?  In both of these instances, there is the potential for significant confounding, but that would be more concerning if the latter pattern applies, because of the possibility that the LM might be used in lesions and there are a priori determined to present less risk.  Any information about the number of anesthesiologists involved, the number of cases by each and the distribution of airway types by anesthesiologist, would be useful in this regard.

We thank the reviewer and have amended the methods section accordingly:

“The decision about the type of airway management (LMA or ET) was made by the supervising pediatric anesthetist based on their experience/opinion with regard to the risk of aspiration associated with the surgical procedure. The airway procedure was pre-determined. Patients with an individual risk for/against one of the two procedures were excluded. There was no supervising anesthetist who used both procedures, i.e. either laryngeal mask or endotracheal tube for all patients under care.”

The results section has been supplemented accordingly: “Final-year-residents or junior specialists performed the airwayprocedure closely supervised by six dedicated pediatric anesthetists, three of whom preferred LMA and three ET for the studied surgical procedure.”

The Discussion should also address the above point in some detail, irrespective of how much data are available on this issue. It is inadequate to state that "this study has all the known limitations".

The authors thank the reviewer for the objection. The manuscript has been revised to emphasize the selection criteria, participating anesthesiologists, and limitations of the retrospective study design.

The discussion section was supplemented: “There is no expert consensus on definitive airway protection during pediatric lacrimal duct stenosis surgery. Even in the author's medical center, there are different approaches. The team of dedicated pediatric anesthetists is actually divided into two cohorts: one part of the group prefers endotracheal intubation, while others are comfortable with the use of a laryngeal mask airway. Under their supervision, the junior fellows performed the different airway procedures.”

The limitations have been specified: “Our study was not without limitation. First, the current study is retrospective and therefore the retrospective design causes some limitations: The accuracy of the retrospective review of the handwritten medical records can be questioned. However, artefacts or true adverse events may be better distinguished or filtered out by manual documentation than by electronic documentation. Second data analysis could be hampered by incomplete data sets. In addition, the evaluated time intervals should be considered as surrogate parameters for the effective induction and recovery time and may be influenced by various variables that cannot be determined retrospectively. Third, the regular pediatric anesthesia stuff performed the procedures, not a single study physician. At the same time, diversity may offer advantages such as practical application and user-independent comparability.”

The conclusion is clear, direct, and justified based on the data presented.

We thank the reviewer.

Please correct misspellings:

In Table 1: “Femal” gender

In Table 3: “Postoperativ”

The manuscript has been revised accordingly:

“Female gender”

“Postoperative pain treatment “

Reviewer 2 Report

Comments and Suggestions for Authors

MS Title: Laryngeal mask airway versus endotracheal intubation during …….

MS Authors: Nicolas Leister et al.

MS Number: Children-2900425

Date of Review: 2024/02/26

In this retrospective case study, the authors tried to evaluate the role of LMA versus endotracheal tube intubation in pediatric patients (younger than 6 years old) undergoing lacrimal duct surgery. The authors concluded that LMA is a safe airway method for infantile lacrimal duct stenosis pediatric patients population.

Comments:

1.    Page 1, line 19: “LM” airways is better to be replaced by “LMA” (which is more commonly used)

2.    Page 1, line 26: “surplus” meaning “residual”? Also, please identify such residual effect of NMBAs in this study by quantitative evidence (e.g., TOF or other objective physical parameters)?

3.    Page 2, line 44: Please define the “signs for aspiration” in the latter text.

4.    Page 2, line 57: “…there is a risk of endotracheal aspiration of irrigation fluid, blood, and/or pus.” Meanwhile, the authors also cited the reference-7 of a role of suction tube placed in patient’s hypopharynx (when LMA be used) to ensure no irrigation fluid went into un-protected trachea. Here, it might be necessary for the authors to explain the rationale of not placing any suction mechanics under such circumstance (especially this has been shown in literature).

5.    Page 2, line 72: Please describe in more details the statistical analysis plan (SAP) to test the primary and secondary outcome parameters in this study.

6.    Page 2, line 76: Please describe the prevalence rate of congenital lacrimal duct stenosis beyond and above 6 years old pediatric population in your hospital.

7.    Page 2, line 83: Please describe in more details the “standardized anesthesia protocols”. This is important because late analysis in this article (e.g., Table 2) involved intra-op/post-op events related to anesthesia depth, emergence, NMBAs effects, vitals (BP, oxygen saturation etc.). Therefore, all the info regarding induction/maintenance/emergence of anesthesia are crucial.

8.    Page 3, line 93: Please describe in more details “the decision on the mode of anesthesia” on such pediatric lacrimal duct surgery in this hospital/. How diverse the habits and decisions of anesthesia staff to select LMA or ET intubation in such pediatric population in this study?

9.    Page 3, line 120: “Chi-square test”, “Mann-Whitney U-test”

10. Page 4, line 136,Table 2: Mis-spelling of “opioid”.

11. Page line 136, Table 2 and Table 3: Please explain the rationale of the expression of “total”.

12. Page 5, line 139: Please describe NSAIDs in more details.

13. Page 5, line 157, Table 5: “SpO2”, not “Sp02)

14. Page 5, line 160: “anesthesia periods” should be “anesthesia duration”

15. Page 5, line 162 and 165: Please describe the event of such “long” induction and emergence duration in these cases.

16. Page 6, line 173: How was the “Residual neuromuscular block” was detected here and in Table 7?

17. Page 6, line 176, Table 7: Should “effect” be “management”? Case 4: If “bronchial obstruction” was the cause of the AE, why could LMA could reverse such event? Cases 6 and 7: please describe “drug therapy”

18. Page 7, line 192: POR stay was longer in LMA group and such difference was attributed to the use of opioids (line 194). Was remifentanil used mostly in this study?

19. Page 7, line 208: The statement of “no relevant aspiration events occurred in the LM cohort,…. ” was the major argument in this paper. Therefore, the definition and validation method of occurrence of aspiration events should be carefully described here.

20. Page 7, line 210: The statement of “Continuous suctioning, as described in the literature does not appear to be necessary to reduce the risk of aspiration” might be arguable here.

21. Page 7, line 218: Regarding the statement: “The time intervals evaluated should be considered as surrogate parameters for the effective induction and recovery time ….”, the authors need to clarify the secondary outcome parameters in this study.

22. Page 7, line 222: The authors made conclusion of “Laryngeal mask airway can be safely used for airway management….”. Please support this notion by providing statistical analysis results.

Author Response

Point-to-Point-Reviewer Response

Dear Editors & Reviewers

Thank you for having reviewed our manuscript. Please find enclosed the revised version of our manuscript entitled “Laryngeal mask airway versus endotracheal intubation during lacrimal duct stenosis surgery in children – a retrospective analysis”.

We are grateful for the reviewer´s invaluable comments and have adjusted the manuscript according to their recommendations. Please find below a point-by-point response, and attached the manuscript according to the editorial instructions. We hope that we have addressed all concerns appropriately and are looking forward to your decision.

Regards,

Nicolas Leister

on behalf of all co-authors

REVIEWER COMMENTS.

Reviewer #2:

In this retrospective case study, the authors tried to evaluate the role of LMA versus endotracheal tube intubation in pediatric patients (younger than 6 years old) undergoing lacrimal duct surgery. The authors concluded that LMA is a safe airway method for infantile lacrimal duct stenosis pediatric patients population.

1.    Page 1, line 19: “LM” airways is better to be replaced by “LMA” (which is more commonly used)

We thank the reviewer and have revised the manuscript accordingly. Changes were made to the entire manuscript.

2.    Page 1, line 26: “surplus” meaning “residual”? Also, please identify such residual effect of NMBAs in this study by quantitative evidence (e.g., TOF or other objective physical parameters)?

We thank the reviewer for his objection and have clarified the statement with regard to the NMBAs as follow:

“Five of these were directly or indirectly related to ET (laryngo-/bronchospasm; muscle relaxant residual)”

Objective parameters were added to Table 7:

“residual neuromuscular block (Train-of-four: 3/4)”

3.    Page 2, line 44: Please define the “signs for aspiration” in the latter text.

We would like to thank the reviewer for this comments and have clarified the presentation accordingly:

Page 2, line 44: “Signs of relevant lower airway aspiration do not appear to occur. “

We have added in the Materials and Methods section: “In particular, evidence of relevant aspiration events was sought: Description in the anesthesia record (e.g. subglottic secretions, gas exchange disturbance, need for excessive ventilatory support or supplemental oxygen), admission to the intensive/intermediate care unit during hospitalization, and readmission after discharge.”

The results have been supplemented as follows:

“No patient had to be transferred to the intensive care unit (all patients could be transferred to the regular ward), nor did any patient have to be readmitted after discharge from the university hospital. All patients left the hospital the day after surgery. “

4.    Page 2, line 57: “…there is a risk of endotracheal aspiration of irrigation fluid, blood, and/or pus.” Meanwhile, the authors also cited the reference-7 of a role of suction tube placed in patient’s hypopharynx (when LMA be used) to ensure no irrigation fluid went into un-protected trachea. Here, it might be necessary for the authors to explain the rationale of not placing any suction mechanics under such circumstance (especially this has been shown in literature).

The authors would like to thank the reviewer for this important objection. The reasons for not using continuous suction are explained in detail: “At the authors´ institution, both endotracheal intubation and laryngeal masks are used for airway management in this surgical procedure; both procedures are used without continuous suctioning, based on the prevailing opinion at the authors´ hospital that this is more likely to cause mucosal lesions and that manipulation of the airway should be minimized in this way. To our knowledge, there is no general recommendation from the German or European society of anesthesiology.”

5.    Page 2, line 72: Please describe in more details the statistical analysis plan (SAP) to test the primary and secondary outcome parameters in this study.

We thank the reviewer and have revised the manuscript accordingly: “We retrospectively identified all children younger than 6 years of age who underwent lacrimal duct stenosis surgery during the corresponding period and analyzed their anesthesia protocols. The primary objective of this study was the analysis of type of airway management (laryngeal mask airway or endotracheal tube). Furthermore, we reviewed for procedure-related complications. In particular, indications of relevant aspiration events were sought: Description in the anesthesia record (e.g. subglottic secretions, gas exchange disturbance, need for excessive  ventilatory support or supplemental oxygen), admission to the intensive/intermediate care unit during hospitalization, and readmission after discharge.

The secondary objective was to analyze the anesthetic approach (induction time: beginning of induction to surgery release by anesthesia team; emergence time: end of surgery to leaving operation room; recovery room time: start recovery room to recovery room discharge; anesthetic medication, and postoperative pain therapy) for probing, dilating, and irrigating the lacrimal duct in children younger than 6 years of age.

6.    Page 2, line 76: Please describe the prevalence rate of congenital lacrimal duct stenosis beyond and above 6 years old pediatric population in your hospital.

To clarify we added to the text: “The prevalence of patients with lacrimal duct stenosis younger than 6 years of age is 95.6% at our university hospital (regarding all patients younger than 18 years). As a tertiary center, we are only assigned cases that are refractory to conservative treatment.”

7.    Page 2, line 83: Please describe in more details the “standardized anesthesia protocols”. This is important because late analysis in this article (e.g., Table 2) involved intra-op/post-op events related to anesthesia depth, emergence, NMBAs effects, vitals (BP, oxygen saturation etc.). Therefore, all the info regarding induction/maintenance/emergence of anesthesia are crucial.

We thank the reviewer and have added an explanation to our documentation: “In the aforementioned anesthesia protocols used throughout the hospital, all analyzed events/vital signs were documented in a timeline and clearly timestamped (e.g., start of recovery room: 10:02; end of recovery room: 11:03; etc.). Adverse events and measures are documented according to established clinical standards.”

8.    Page 3, line 93: Please describe in more details “the decision on the mode of anesthesia” on such pediatric lacrimal duct surgery in this hospital/. How diverse the habits and decisions of anesthesia staff to select LMA or ET intubation in such pediatric population in this study?

We thank the reviewer for his advice and have clarified the decision-making process accordingly: “The decision about the type of airway management (LMA or ET) was made by the supervising pediatric anesthetist based on their experience/opinion with regard to the risk of aspiration associated with the surgical procedure. The airway procedure was pre-determined. Patients with an individual risk for/against one of the two procedures were excluded. There was no supervising anesthetist who used both procedures, i.e. either laryngeal mask or endotracheal tube for all patients under care “

9.    Page 3, line 120: “Chi-square test”, “Mann-Whitney U-test”

The manuscript has been revised accordingly: “Chi-square test and Mann-Whitney U-test”

10. Page 4, line 136, Table 2: Mis-spelling of “opioid”.

We thank the reviewer and have revised the manuscript at line 136, Table 2: “Opioid“

11. Page line 136, Table 2 and Table 3: Please explain the rationale of the expression of “total”.

The authors would like to thank the reviewer for this important objection. We agree that this term was misleading and not consistently used. “Total” has been removed and we have adjusted the percentages (proportion relative to the cohort) in Table 2 and Table 3.

12. Page 5, line 139: Please describe NSAIDs in more details.

We thank the reviewer and have named the NSAIDs used: “ibuprofen or paracetamol”

13. Page 5, line 157, Table 5: “SpO2”, not “Sp02

The reviewer is right and we have revised the manuscript accordingly: “SpO2

14. Page 5, line 160: “anesthesia periods” should be “anesthesia duration”

We have changed periods to duration as suggested.  “Anesthesia duration”

15. Page 5, line 162 and 165: Please describe the event of such “long” induction and emergence duration in these cases.

We thank the reviewer for this objection and have clarified the information provided: “…although one excessively long induction (duration 45 minutes due to difficult venous access) in cohort ET was excluded from the analysis. Duration from end of surgery to start of recovery room stay was significantly shorter in cohort LMA (p=0.03), although two excessively long periods (41 and 45 minutes, without documented reason) in cohort ET were excluded from the analysis…”

16. Page 6, line 173: How was the “Residual neuromuscular block” was detected here and in Table 7?

We thank the reviewer. We have clarified the objectification of neuromuscular block: “Residual neuromuscular block (Train-of-four (TOF) 3/4 at the end of surgery) after administration of muscle relaxants to induce anesthesia also occurred in the ET cohort”.

Table 3 has been revised accordingly:  “(TOF 3/4)”

17. Page 6, line 176, Table 7: Should “effect” be “management”? Case 4: If “bronchial obstruction” was the cause of the AE, why could LMA could reverse such event? Cases 6 and 7: please describe “drug therapy”

We thank the reviewer and have supplemented Table 7.

1. “Effect” has been replaced by “management“.

2. The description of the management of Case 4 has been expanded: “beta adrenergic drug application, repeated propofol administration, ET removal and LM insertion, prolonged wake-up”.

3. Case 6: “propofol administration in recovery room”

4. Case 7: “propofol administration in recovery room”.

18. Page 7, line 192: POR stay was longer in LMA group and such difference was attributed to the use of opioids (line 194). Was remifentanil used mostly in this study?

We thank the reviewer for this objection. Remifentanil was used in 76 of the 84 patients (see table 2). Prolonged POR stay could be related to the use of postoperative opioids (see table 3).

We have added to the text: “…particularly the significantly more frequent use of postoperative opioids…”

19. Page 7, line 208: The statement of “no relevant aspiration events occurred in the LM cohort,…. ” was the major argument in this paper. Therefore, the definition and validation method of occurrence of aspiration events should be carefully described here.

We are grateful to the reviewer´s valuable objection. We have clarified the criticism in order to identify any aspiration events.

We have added in the Material and Methods section:

“In particular, indications of relevant aspiration events were sought: Description in the anesthesia record (e.g. subglottic secretions, gas exchange disturbance, need for excessive ventilatory support or supplemental oxygen), admission to the intensive/intermediate care unit during hospitalization, and readmission after discharge.”

At page 7 line 208 we have supplemented as follows:

“Overall, the data presented here indicate that no relevant aspiration events (no description of any aspiration event in the anesthesia record (e.g. subglottic secretions), no gas exchange disturbance or need for ventilatory support, supplemental oxygen or admission to the intensive/intermediate care unit during hospitalization, no readmission after discharge) occurred in the LMA cohort…..”

20. Page 7, line 210: The statement of “Continuous suctioning, as described in the literature does not appear to be necessary to reduce the risk of aspiration” might be arguable here.

We thank the reviewer and have added to the text as follows: “Continuous suctioning, as described in the literature does not appear to be necessary to reduce the risk of aspiration. In this way, the potential risk of mucosal lacerations from continuous upper airway manipulation could be reduced.”

21. Page 7, line 218: Regarding the statement: “The time intervals evaluated should be considered as surrogate parameters for the effective induction and recovery time ….”, the authors need to clarify the secondary outcome parameters in this study.

We thank the reviewer and have added to the Materials and Methods section:

“The secondary objective was to analyze the anesthetic approach (induction time: beginning of induction to surgery release by anesthesia team; emergence time: end of surgery to leaving operation room; recovery room time: start recovery room to recovery room discharge…”

22. Page 7, line 222: The authors made conclusion of “Laryngeal mask airway can be safely used for airway management….”. Please support this notion by providing statistical analysis results.

We thank the reviewer and have revised page 7, line 222:

“Anesthesia induction (p=0.006) and emergence (p=0.03) were faster using LMA compared to ET. There was no evidence of increased risk of aspiration in those patients treated with LMA. Thus, LMA can be safely used for airway management in pediatric patients younger than 6 years for lacrimal duct surgery including probing, dilating, and irrigating of the lacrimal duct.

Round 2

Reviewer 2 Report

Comments and Suggestions for Authors

Dear authors

Thanks for your replies and discussion on my comments on your submitted manuscript. Great job.